# Dye-incorporated coordination polymers for direct photocatalytic trifluoromethylation of aromatics at metabolically susceptible positions

Tiexin Zhang[1], Xiangyang Guo[1], Yusheng Shi[1], Cheng He[1] & Chunying Duan [1,2]

Direct trifluoromethylation of unactivated aromatic rings at metabolically susceptible positions is highly desirable in pharmaceutical applications. By incorporating thiophenes into the backbone of triphenylamine to enlarge its π-system, a new approach for constructing coordination polymers is reported for direct trifluoromethylation without prefunctionalization of the aryl precursors. The improved light-harvesting ability and well-modulated excited state redox potential of the designed polymers endow the generated CF$_3$ radicals with suitable reactivity and enhance radical adduct oxidation in pores. The well-configured interactions between the organic ligands distort the coordination geometry to create active interaction sites within the coordination polymer; thus, the substrates could be docked near the photoredox-active centres. The synergistic electronic and spatial effects in the confined pores balance the contradictory demands of electronic effects and reaction dynamics, achieving regio- and diastereoselective discrimination among reaction sites with unremarkable electronic/steric differences.

[1] State Key Laboratory of Fine Chemicals, Dalian University of Technology, Dalian 116024, China. [2] Collaborative Innovation Center of Chemical Science and Engineering, Tianjin 300071, China. Correspondence and requests for materials should be addressed to C.D. (email: cyduan@dlut.edu.cn)

The installation of a trifluoromethyl group onto metabolically susceptible positions on aromatic rings increases the robustness of the corresponding drug candidates towards oxidative metabolism in vivo[1], making the trifluoromethylation of aromatics a highly desirable chemical reaction[2,3]. In light of the photoinduced generation of highly reactive $CF_3$ radicals, polypyridyl noble metal complexes enable direct C–H trifluoromethylation of unactivated aromatics under visible-light irradiation[4]. This approach bypasses tedious prefunctionalization of aryl precursors that is needed for transition metal-catalysed cross-couplings using a nucleophilic or electrophilic source of $CF_3$[5,6]. However, the discrimination of various aromatic positions without distinct electronic contributions is thermodynamically difficult due to the high electrophilicity of $CF_3$ radicals. Trifluoromethylation of drugs with aromatic cores can protect them from aryl oxidative metabolism by enzymes in living cells (i.e., cytochrome P450). Thus, appending trifluoromethyl groups to the metabolically susceptible positions of aromatics without prefunctionalization of precursors[7] is highly desirable, and it is promising to achieve this goal by merging all the catalytic requirements to controllably generate $CF_3$ radicals under visible light and trigger the metabolically susceptible positions of aromatic drug candidates.

Porous coordination polymers exhibit confined pores similar to those in molecular sieves[8,9], and the tenability and flexibility of the material allow for uniform heterogenization of photoactive organic dyes at high densities around the pores[10–14]. As the integration of near-ultraviolet (UV) responsive triphenylamine (TPA) and covalently binding group L-proline derivative into a single coordination polymer has been proven to be an effective approach to the photocatalytic α-[15] or β-functionalization[16] of saturated carbonyl compounds, we believe that merging the TPA-based chromophore with modulated photoelectronic property and the noncovalently binding site together within one framework is a potential strategy for controllably generating $CF_3$ radicals under visible-light irradiation and harnessing the aromatic drug candidates that lack of covalent binding moieties.

Here, we show that incorporating thiophene groups into the ligand[14] bearing TPA moiety allows us to modify the potential photochemical properties of TPA-based coordination polymer with the aim of formulating a new system for direct trifluoromethylation of aromatics in a heterogeneous manner (Fig. 1). We envision that the enlarged π-system created by the insertion of thiophene moieties into the backbone of the TPA-based ligand would improve the visible-light harvesting ability and tune the photoelectronic performance to generate reactive $CF_3$ radicals with milder reactivity, and allow for more efficient oxidation of the radical adduct. Simultaneously, enhanced interactions between organic dyes cause the metal nodes to distort, creating coordination vacancies for the docking of substrates in close proximity to the photocatalytically active centres (Fig. 1c). Furthermore, the intrinsic crystalline nature of coordination polymers allows for structural analyses of the intermediates formed in each activation and catalytic step, offering an excellent platform for studying photon capture, electron delivery, and catalytic activation[17,18].

## Results

### Synthesis and characterization of the photocatalyst. 
A solvothermal reaction between $H_3$TCTA (tris[4-(5-carboxy-2-thienyl)phenyl]amine) and $Zn(NO_3)_2 \cdot 6H_2O$ in DMF at 100 °C afforded a new coordination polymer, Zn–TCTA, in a 70% yield. Single-crystal X-ray structural analysis revealed that this polymer crystallizes in the $R$-$3c$ space group. The distorted geometry of the $Zn_4O$ units is enforced by interlayer π···π interactions between

TCTA moieties, endowing zinc atoms with empty coordination sites for potential substrate binding and activation (Supplementary Fig. 4). Each $TCTA^{3-}$ anion bridges three $Zn_4O$ clusters, and each $Zn_4O$ cluster is connected to six different $TCTA^{3-}$ anions, forming a two-dimensional sheet. These deprotonated ligands are located above or below the two-dimensional sheets, creating undulating surfaces and intralayer cavities in the metal–organic sheets (Fig. 1b). Adjacent sheets assemble in an ABCABC fashion through π···π stacking interactions between the planar 4-thiophenylphenyl moieties of the ligands to form a three-dimensional porous structure (Fig. 1d). It is postulated that the interlayer stacking mode can enhance the excitation delocalization and light-harvesting ability of polymers[19], benefiting photocatalytic reactions under visible-light irradiation. Additionally, the interlayer stacking mode is also expected to provide the possibility of exfoliating the polymers into two-dimensional materials[20], which is expected to enhance their catalytic properties. The hexagonal pocket windows of the intralayer cavities were plugged from the direction of the $c$-axis to retain horizontal open channels with cross-sections of $10.4 \times 18.0$ Å$^2$ (Fig. 1d). A methylene blue dye uptake experiment with Zn–TCTA was used to demonstrate the guest accessibility, and the experiement yielded a dye uptake of 2.8% of the catalyst weight, as determined by the UV–vis spectra (Supplementary Fig. 8).

The UV–vis absorption spectrum of solid-state Zn–TCTA exhibited a remarkable redshift and broader absorption band than that observed for TPA-based MOF–150[14] (Fig. 2a and Supplementary Fig. 9). Solid-state electrochemical measurements revealed an oxidative potential of 1.17 V (Supplementary Fig. 10), and the reductive potential of the excited state was determined to be −1.22 V on the basis of a free energy change of 2.39 eV (Fig. 2a). In comparison to the photocatalyst $fac$-Ir(Fppy)$_3$[21], Zn–TCTA has a weaker but sufficient photoreductive ability to reduce trifluoromethanesulfonyl chloride (TfCl, $E_{1/2}^{red} = -0.18$ V vs. saturated calomel electrode)[4] and generate $CF_3$ radicals with milder reactivity. The more positive oxidative potential of the dyes in Zn–TCTA is expected to oxidize the $CF_3$ radical adduct more efficiently to complete the catalytic cycle. The quenched emission at 546 nm (Fig. 2b) and the substantially decreased fluorescence lifetime (Supplementary Fig. 12) of the Zn–TCTA suspension upon the addition of TfCl indicated a classical photoinduced electron transfer process from the excited state of the TCTA moiety to TfCl, which directly yielded active $CF_3$ radicals.

### Heterogeneous photocatalytic trifluoromethylation. 
In a typical procedure, a reaction mixture of heterocycle 1a (0.25 mmol), 2,4,6-collidine (0.50 mmol), TfCl (0.50 mmol), and 2.5 mol% Zn–TCTA (per mole organic dye ligand) in acetonitrile (1 mL) was subjected to visible-light irradiation from a 23 W household light bulb. An 84% yield of the desired product 2a with the $CF_3$ group positioned at the metabolically susceptible α-position of the carbonyl group[22] was obtained after 24 h (Figs. 3c, and 5, 2a). The control experiments indicated that light, an inert atmosphere, and Zn–TCTA were indispensable for the reaction to proceed (Supplementary Table 2, entries 5–7). It should be noted that the addition of the radical scavenger tetramethylpiperidine-$N$-oxyl (TEMPO) to the reaction mixture inhibited the reaction (Supplementary Table 2, entry 13). For the reaction mixture with 2-methyl-2-nitrosopropane dimer (MNP dimer, a typical $CF_3$ radical scavenger) as a substitute for the substrate, a singlet–triplet splitting signal corresponding to the $CF_3$–MNP• adduct was observed by in situ electron paramagnetic resonance (EPR) spectroscopy ($g$ value = 2.006)[23] under visible-light irradiation. These results suggested the formation of $CF_3$ radicals via

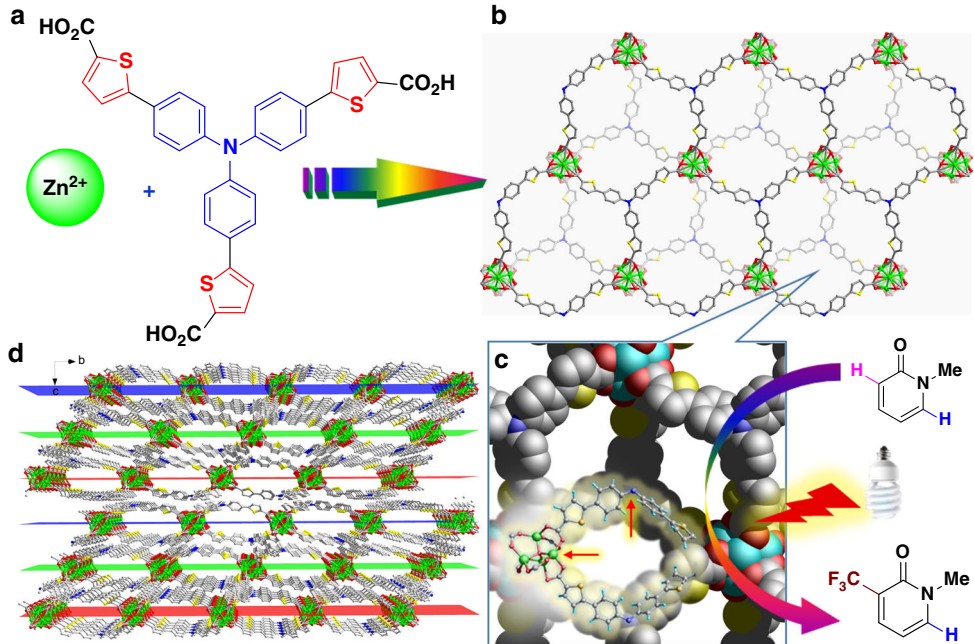

**Fig. 1** Schematic illustration of the composition and structure of Zn–**TCTA**. **a** The components. **b** The top view of the undulating monolayer with a virtual plane. **c** The intralayer cage showing the potential substrate binding site. **d** The packing pattern and open channels along the *a*- or *b*-axis. Green, Zn; yellow, S; red, O; blue, N; gray, C. Hydrogen atoms and solvent molecules are omitted for clarity

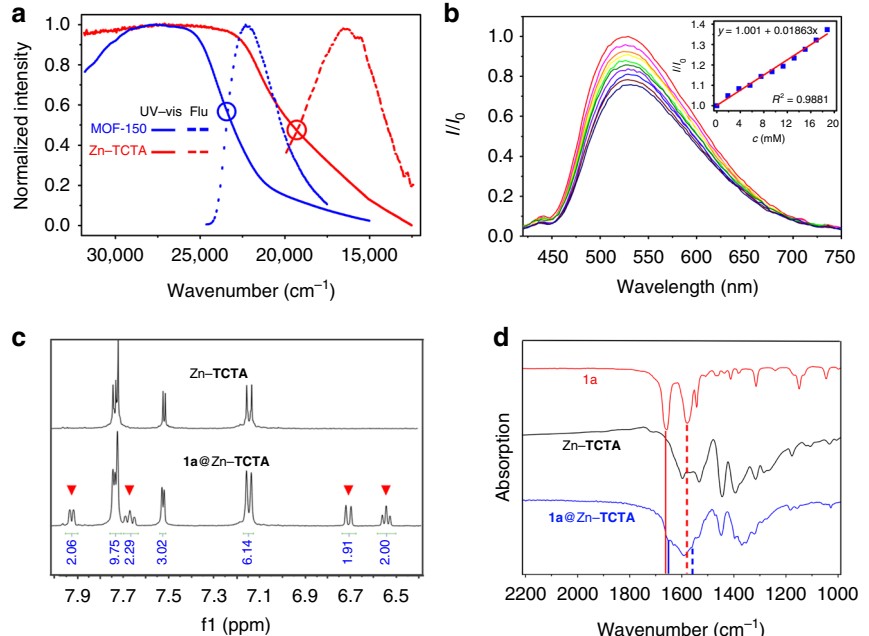

**Fig. 2** The host–guest interactions between Zn–**TCTA** and substrate/reagent. **a** Normalized absorption and emission spectra of Zn–**TCTA** ($\nu^{0-0} = 19{,}274$ cm$^{-1}$, $E^{0-0} = 2.39$ eV) and MOF–150 ($\nu^{0-0} = 23{,}468$ cm$^{-1}$, $E^{0-0} = 2.91$ eV) excited at 390 and 350 nm, respectively. **b** Fluorescence spectrum of Zn–**TCTA** upon the addition of trifluoromethanesulfonyl chloride (TfCl) and the corresponding simulated Stern–Volmer curve (inset of **b**) excited at 390 nm; the intensity was recorded at 546 nm. **c** $^1$H NMR spectrum of the crystals of Zn–**TCTA** and **1a**@Zn–**TCTA** (digested in DMSO-$d_6$/DCl). Peaks marked with inverted red triangles represent the aromatic signals of the encapsulated substrate **1a**. **d** Comparison of the infrared (IR) spectrum of **1a** (red line), Zn–**TCTA** (black line), and **1a**@Zn–**TCTA** (blue line)

a photoinduced single electron transfer pathway (Supplementary Fig. 29). $^1$H nuclear magnetic resonance (NMR) analysis of the digested crystals revealed that Zn–**TCTA** could adsorb approximately 2.0 equivalents (equiv) of **1a** per unit (Fig. 2c). The infrared (IR) spectrum of the **1a**-impregnated crystals showed the C=C

stretching vibrations of **1a** at 1560 cm$^{-1}$, indicating the fixation and possible activation of the alkene moiety within the pores of the coordination polymer (Fig. 2d).

Single-crystal X-ray structural analysis on the substrate-impregnated crystal **1a**@ Zn–**TCTA** revealed that the framework

and pores of Zn–**TCTA** were retained upon uptake of the substrate[18,24], and indicated that **1a** was fixed near the inner surface of the confined pores by coordination interactions between the unprotected zinc atom and the carbonyl oxygen of **1a** ($O_{1a}$···Zn distance of ca. 2.20 Å). The enforced close proximity between the photoactive **TCTA**$^{3-}$ unit and the metabolically susceptible $\alpha$-position of **1a** (shortest interatomic distance of ca. 3.75 Å for $\alpha$-position vs. ca. 4.11 Å for $\delta$-position, respectively) provided the possibility to accelerate the reaction at the $\alpha$-position[16] (Figs. 3d, e and Supplementary Fig. 18). Because the oxidation of the CF$_3$ radical adduct, which is known to be the rate-determining step[6], could be accelerated by the closer spatial proximity in the case of CF$_3$ radical addition at the $\alpha$-position of the carbonyl group, the specific spatial orientation of Zn–**TCTA** in the pores is expected to create the distinctive regioselectivity of trifluoromethylation in comparison to photocatalysis at the $\delta$-position by $fac$-Ir(Fppy)$_3$[4] (Fig. 3b). In this case, both the electronic and spatial effects are important to the heterogeneous reaction in the pores of Zn–**TCTA** with microdynamics that are distinct from those of the homogeneous reaction.

After photocatalysis, Zn–**TCTA** was easily isolated from the reaction mixture by centrifugation, and the time-dependent conversion plots revealed that the photocatalyst could be reused at least three times without a remarkable decrease in reactivity (Fig. 4c and Supplementary Table 2, entries 10–12). The unchanged substrate ingress/egress ability of the recovered solids after three runs compared with that of the Zn–**TCTA** block crystals suggested that the capacity of the Zn–**TCTA** pores was mainly retained (Supplementary Fig. 14). The X-ray powder diffraction pattern of the recovered catalyst indicated that the integrity of Zn–**TCTA** was basically maintained during the reaction. The remarkably decreased diffraction peak observed at a low angle possibly suggested the deterioration of long-range order in the vertical direction (Supplementary Fig. 25). It should also be noted that the micron-sized thin layers were exfoliated[25,26] in situ from single-crystal blocks of the polymer on the basis of the scanning electron microscopy (SEM) (Supplementary Fig. 26) and transmission electron microscopy (TEM) (Fig. 4d and Supplementary Fig. 27) images. Low-pressure N$_2$ adsorption/desorption of fresh Zn–**TCTA** and the recycled sample were conducted at 77 K. Both of fresh crystals of Zn–**TCTA** and the recovered sample exhibited a Type-I sorption behaviour, with Brunauer–Emmett–Teller (BET) surface areas of 1383 and 1600 m$^2$ g$^{-1}$, respectively, which were calculated from N$_2$ adsorption isotherms. Compared with the case of fresh Zn–**TCTA**, the mildly increased BET surface area of recovered sample might be attributed to the in situ exfoliation. The maintained pore size distributions reflected the stability of coordination polymer and the integrity of pores during photocatalysis (Supplementary Fig. 28).

Considering the unavoidable loss when recycling a small amount of catalyst after each run, a time-course experiment was performed using a large excess of **1a** (2.5 mmol, 10 equiv). With the same quantity of Zn–**TCTA** (6.25 µmol, 2.5 mol%) and intermittent charging of the original amounts (0.50 mmol, 2.0 equiv) of TfCl and the base additive each day, a final NMR yield of ca. 95% was obtained after ten consecutive catalytic runs without further derivatization of **2a**. It should be noted that the homogeneous control experiment exhibited remarkably destructive consumption of product **2a'** when exposed to an excess of the reactive species over prolonged reaction times (Fig. 4b).

This approach can be applied to aromatic substrates bearing common heteroatomic functional groups that may act as drug candidates, enabling the photocatalytic reaction without the need for prefunctionalization of the arenes (Fig. 5). Gratifyingly, bioactive heteroarenes, such as derivatives of uracil (**1b**), caffeine

(**1d**), theophylline (**1e**), and coumarin (**1f**), can be trifluoromethylated at metabolically susceptible positions with comparable yields. Note that this methodology was successfully extended to a series of fine chemicals or drug candidates with ambiguous aromatic reaction sites, such as the flavourant methylvanillin (**1l**), the female hormone estrone derivative (**1m**), and the nonsteroidal antiinflammatory drug derivatives of ibuprofen (**1n**) and indomethacin (**1o**) with distinctive regioselectivity in comparison to the case using $fac$-Ir(Fppy)$_3$ as a photocatalyst (Supplementary Fig. 21). When substrate **1p**, which has a molecular size larger than the cross-section of the open channel in the pores of Zn–**TCTA** (Supplementary Fig. 16), was employed, a lower yield (<10%) was detected (**2p**). Such a size-dependent transformation suggests that the catalytic reaction occurs within the pores of Zn–**TCTA**, and the specific spatial orientation in the pores of Zn–**TCTA** can forge the improved and distinctive regioselectivity of trifluoromethylation compared with that obtained via homogeneous photocatalysis (Fig. 3b)[4]. Moreover, this heterogeneous route was amenable to either gram-scale preparation (**2a**†) or the introduction of a different perfluoroalkyl group by using C$_4$F$_9$SO$_2$Cl (**2aa**) instead of TfCl. In this case, our approach offers a potentially applicable route to effective protection of metabolically labile sites of late-stage drug intermediates via regioselective aryl trifluoromethyl functionalization.

**Heterogeneous photocatalytic trifluoromethylation-arylation.** By employing a series of N-aryl-methacrylamides and N-aroyl-methacrylamides as substrates, we extended the photocatalytic approach to the selective conversion of unsaturated olefin[27] moieties through tandem alkenyl trifluoromethylation-arylation (Fig. 6)[28]. This reaction involves an initial CF$_3$ radical addition to an unsaturated alkene, intramolecular arylation of the CF$_3$ radical adduct[29,30], subsequent oxidation, and final deprotonation to afford biologically interesting 2,2,2-trifluoroethyl oxindoles[31,32] and isoquinolinediones[33] in considerable yields. When substrates bearing *meta*-substituted aryl moieties were used, only one type of regioisomer was obtained, indicating the high regioselectivities of the reactions (**4c–4e**). In addition to CF$_3$, other perfluoroalkyl moieties, such as C$_4$F$_9$, could also be introduced to the oxindole scaffold through this methodology (**4aa**).

Single-crystal structural analysis of **3a**@Zn–**TCTA** suggested that the polymer structure was retained upon uptake of the substrate. Weak coordination interactions were also found between the metal node and carbonyl group of **3a** (O···Zn distance of ca. 3.76 Å), and C–H···$\pi$ interactions were observed between the phenyl edge of **TCTA**$^{3-}$ and phenyl $\pi$-plane of **3a** (Fig. 7a and Supplementary Fig. 19). These interactions docked the substrates in confined spaces to facilitate tandem radical reactions. When no covalent anchoring positions are available on the substrate, the noncovalent substrate–catalyst interactions are critical for stereocontrol of the reaction; however, they are inherently labile toward thermal motions in solutions. Therefore, homogeneous stereoselective photoredox catalysis has long been hindered by the dearth of effective noncovalent binding strategies[34–37]. Inspired by previously described substrate–catalyst interaction patterns, we envisioned that multiple noncovalent interactions within a framework might induce a stereobias in CF$_3$ radical transformations, increasing the molecular robustness towards oxidative metabolism of sp$^3$ C–H bonds with stereospecificity.

As shown in Fig. 6 (i.e., **4f–4j**), N-aroyl-methacrylamide substrates bearing vicinal dialkyl-substituted unsaturated alkene moieties afforded products in good yields and diastereoselectivities as high as 95:5, which are superior to that obtained from a previously reported homogeneous approach[38]. Notably, **3j**,

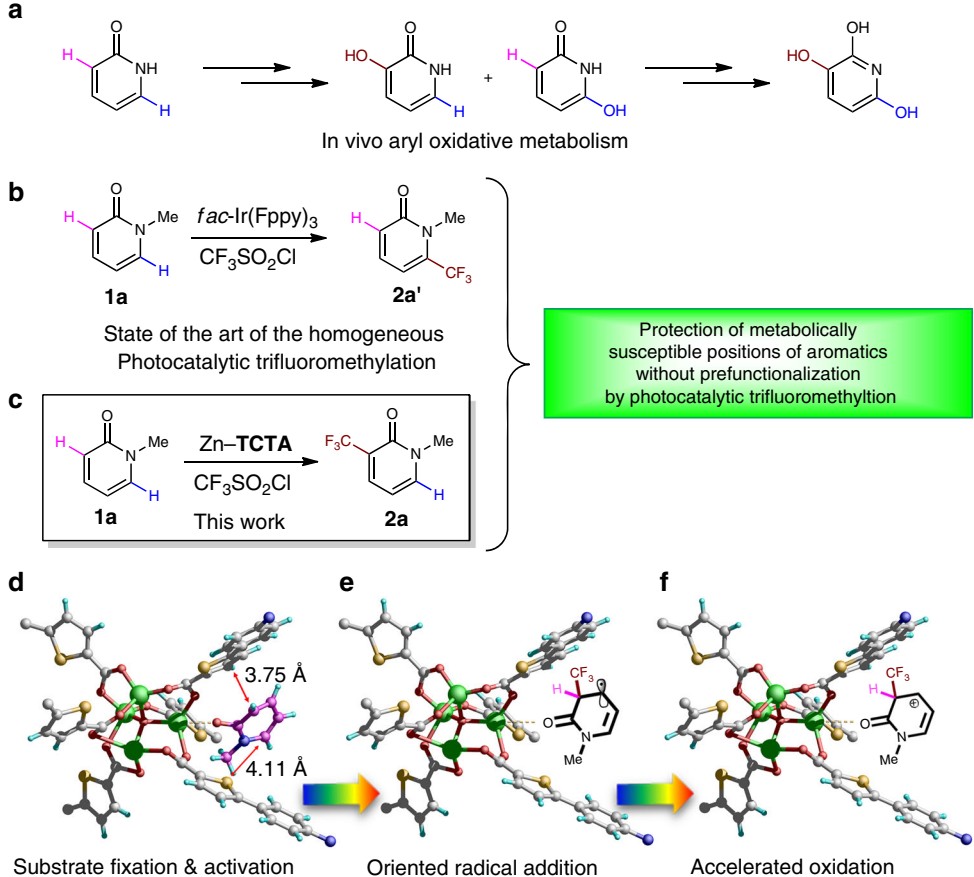

**Fig. 3** Schematic illustration of the photocatalytic site-specific trifluoromethylaion. **a** Potential reaction paths of the in vivo enzymatic aryl oxidative metabolism of (2H)-pyridone. Complementary approaches to the protection of metabolically susceptible sites by trifluoromethylation using **b** *fac*-Ir(Fppy)₃ or **c** Zn–**TCTA** as the photocatalyst, respectively. **d** Crystal structure of **1a**@Zn–**TCTA** showing the multiple interactions between **1a** and the Zn–**TCTA** scaffold and the potential reaction paths **e** and **f** for the reaction

bearing a cyclic allyl moiety, which predominantly forms oligomers in the homogeneous case, was readily converted in a diastereocontrolled manner (d.r. of ca. 94:6) with an acceptable conversion in the presence of Zn–**TCTA**.

A detailed structural analysis of a single crystal of the substrate-enriched framework **3f**@Zn–**TCTA** indicated that within the confined environment, the unsaturated alkenyl and phenyl groups of **3f** were folded in a face-to-face configuration by the C–H···O interactions between the node and *N*-methyl group of **3f** (C···O distance of ca. 3.46 Å) as well as by the C–H···π interactions between the phenyl edge of **TCTA**³⁻ and the phenyl π-system of **3f**. The proximity between the α-carbon atom of the alkenyl group and the hydrogen atom in the *ortho*-position of the phenyl ring shielded the apparent *Re*-face of the alkenyl β-carbon, facilitating the formation of a properly oriented early transition state[39] to direct the tandem radical reactions in a highly diastereoselective manner (Figs. 7b, c and Supplementary Fig. 20).

## Discussion

In conclusion, a new approach was developed for heterogeneous, efficient and site-specific trifluoromethylation of aromatic drug candidates without any directing groups. This approach included the insertion of thiophene moieties into the backbone of a **TPA**-based ligand to extend the light-harvesting ability and tune the photoelectronic properties. The enlarged π-system of the ligand favoured the interaction patterns to distort the coordination geometries of the metal nodes. The potential interactions of the metal nods docked the substrates near the photocatalytically active centres to directly activate the reaction sites of the substrates in special conformations. The comprehensive improvement in the electronic and spatial effects within the coordination polymer balanced the contradictory demands of the electronic effects and reaction dynamics, achieving regio- and diastereoselective discrimination between the reaction sites with unremarkable electronic/steric differences.

## Methods

**Materials and measurements**. Unless otherwise stated, solvents were dried and distilled prior to use according to standard methods. 1,1′,1″-[Nitrilotris(4,1-phenylenethiene-5,2-diyl)]triethanone, the precursor of H₃**TCTA**[40], and the starting materials **1p**[41], **3a** and **3c**[42], **3b**, **3d** to **3i**[43], and **3j**[38] were synthesized according to literature methods. The other substrates were commercially available and used as received. Thin-layer chromatography was carried out on SiO₂ (silica gel 60 F254, Merck), and the spots were identified with UV light.

NMR spectrum were measured on a Bruker Avance 500 WB and Bruker Avance 400 WB spectrometer, and chemical shifts were recorded in parts per million (ppm, δ). High-resolution mass spectrum (HRMS) were recorded on an liquid chromatography/quadrupole–time-of-flight mass spectrometer (Micromass, England) equipped with a Z-spray ionization source. Elemental analyses were performed on a Vario EL III elemental analyser. Inductively coupled plasma (ICP) was conducted on a NexION 300D spectrometer. Thermogravimetric analysis (TGA) was carried out with a Mettler-Toledo TGA/SDTA851 instrument.

Powder X-ray diffractogram measurements were performed with a PANalytical Empyrean X-ray powder diffractometer (Cu Kα radiation, 40 kV, 40 mA). Fourier Transform-IR spectra were recorded using KBr pellets on a JASCO FT/IR-430 instrument. The solid and liquid UV–vis spectra were recorded on a Hitachi U-4100 UV–vis–NIR spectrophotometer and TU-1900 spectrophotometer, respectively. TEM images were collected on a Tecnai F30 microscope. SEM images were carried out over a Nova NanoSEM 450 microscope.

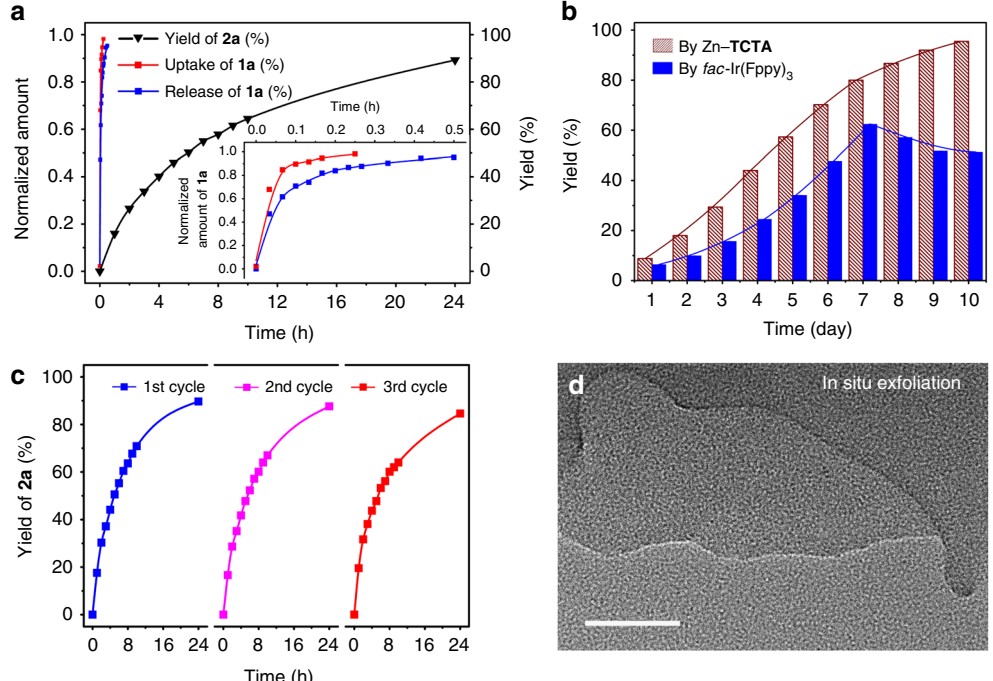

**Fig. 4** Characterization of reaction dynamics and photocatalyst recyclability. **a** The kinetics of the ingress and egress of substrate **1a** on Zn–**TCTA** block crystals in an acetonitrile suspension and the time-dependent conversion plots of the photocatalytic transformation with the insert showing the enlarged ingress/egress curves. **b** Histograms of the time-course reactions of excess **1a** (2.5 mmol, 10 equiv) in the presence of Zn–**TCTA** and *fac*-Ir(Fppy)$_3$ (6.25 μmol, 2.5 mol%) as photocatalysts, respectively. **c** Time-conversion plots of three rounds of the reaction using recycled catalyst Zn–**TCTA**. **d** A magnified TEM image of a crystalline powder of the photocatalyst Zn–**TCTA** after three rounds of reaction, showing the step-like cross-section of the laminated thin layers (inset, scale bar, 20 nm)

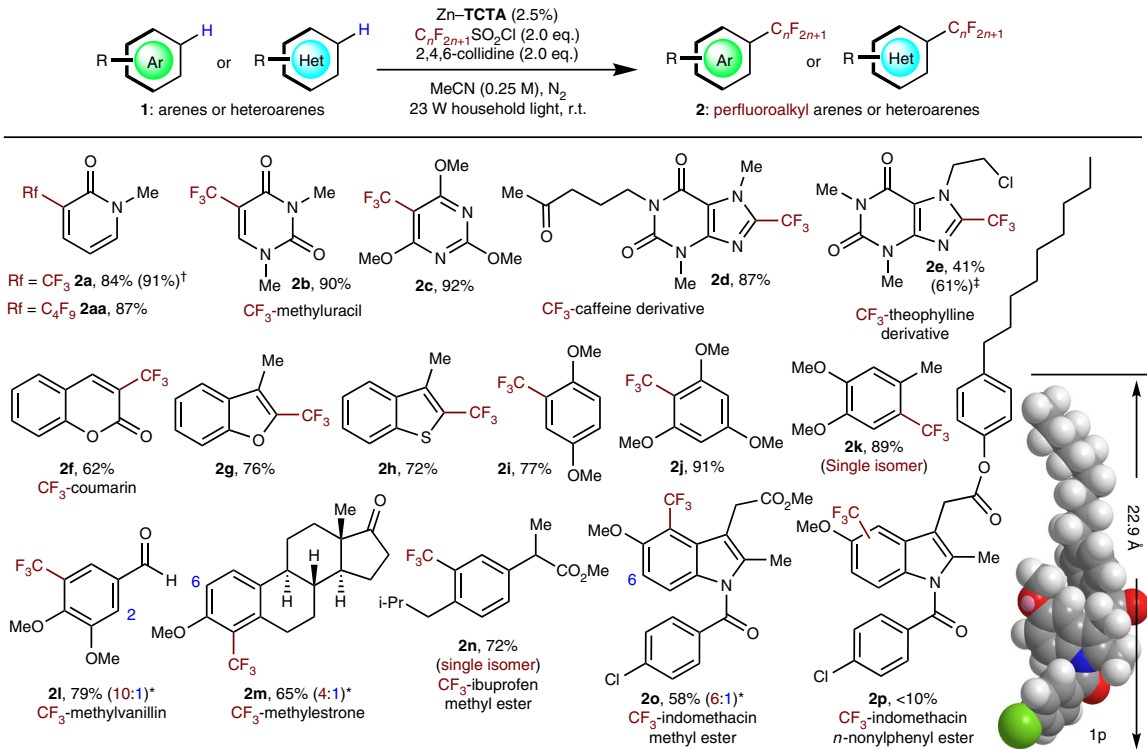

**Fig. 5** Scope of photocatalytic trifluoromethylation of aromatics by Zn–**TCTA**. Reaction conditions: **1** (0.25 mmol, 1.0 equiv), RfCl (fluoroalkylation reagent as specified, 2.0 equiv), base additive (2.0 equiv), Zn–**TCTA** (0.025 equiv), MeCN (1 mL), 23 W household light, N$_2$ atmosphere, room temperature, 24 h. Isolated yields. †15 mmol-scale reaction. ‡3.0 equiv of TfCl and collidine. *Only major regioisomers are shown. The minor regioisomeric sites are labelled with their carbon atom numbers

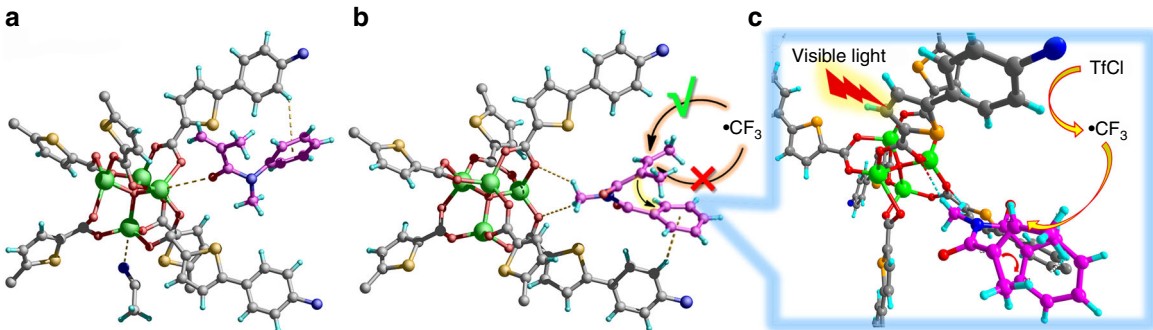

**Fig. 6** Photocatalytic alkenyl trifluoromethylation-arylation by Zn–**TCTA**. Conditions were the same as those in Fig. 5. Isolated yield. †1.5 or ‡3.0 equiv of TfCl and collidine

**Fig. 7** Illustration of the interactions between the encapsulated substrates and Zn–**TCTA**. **a** Structure of **3a**@Zn–**TCTA**. **b** Structure of **3f**@Zn–**TCTA**. **c** The plausible mechanistic interpretation of the role of substrate encapsulation in diastereocontrol

Solid-state cyclic voltammograms were measured using a carbon-paste working electrode; a well-ground mixture of each bulk sample and carbon paste (graphite and mineral oil) was set in the channel of a glass tube and connected to a copper wire. A platinum-wire counter electrode and an Ag/AgCl reference electrode were used in an aqueous solution of KNO₃. The solid fluorescent spectrum were measured on an Edinburgh FS920 instrument. The time-resolved luminescence spectrum were measured on an Edinburgh FLS920 spectrometer.

**Single-crystal X-ray crystallography**. Intensities were collected on a Bruker SMART APEX CCD diffractometer with graphite monochromated Mo-Kα radiation ($\lambda = 0.71073$ Å) using the SMART and SAINT programs[44,45]. The structure was solved by direct methods and refined on F2 by full-matrix least-squares methods with SHELXTL version 5.1[46]. Hydrogen atoms were fixed geometrically at calculated positions and allowed to ride on the parent non-hydrogen atoms. The SQUEEZE program was carried out for crystals Zn–**TCTA**, **1a**@Zn–**TCTA**, and **3a**@Zn–**TCTA**. Crystallographic data for Zn–**TCTA**, **1a**@Zn–**TCTA**, **3a**@Zn–**TCTA**, and **3f**@Zn–**TCTA** are summarized in Supplementary Table 1. For all of the crystal data, one of the carboxylate oxygen atoms was disordered into two parts with the site occupancy factor (s.o.f.) of each part fixed at 0.5.

In the refinement of crystal data of **1a**@Zn–**TCTA**, **3a**@Zn–**TCTA**, and **3f**@Zn–**TCTA**, to help the stability of the refinement for the impregnated related substrate molecules, the bond distances between several atoms were fixed; the geometrical constraints of idealized regular polygons were used for benzene rings, and thermal parameters on adjacent atoms in two molecules were restrained to be similar. The A alert error in the checklist for **3f**@Zn–**TCTA** is due to the partial occupancy of the substrate molecules.

**Synthesis of Zn-TCTA**. A mixture of H₃**TCTA** (93 mg, 0.15 mmol), Zn (NO₃)₂·6H₂O (297 mg, 1.0 mmol) was dissolved in 6 mL DMF in a Teflon-lined steel autoclave. The resulting mixture was kept in an oven at 100 °C for 3 days. The block red-brown crystals for X-ray structural analysis were collected by filtration, washed with acetonitrile, and then dried under vacuum. Yield: 70%. ¹H NMR (400 MHz, DMSO-$d_6$/DCl): $\delta = 7.72$–7.70 (m, 9H), 7.50 (d, $J = 3.9$ Hz, 3H), 7.13 (d, $J = 8.7$ Hz, 3H).

**Dye uptake experiments**. Crystals of Zn–**TCTA** were soaked in a saturated solution of methylene blue in acetonitrile for 12 h, and the resulting crystals were washed with acetonitrile thoroughly until the solution became clear. The dried sample was dissociated by concentrated hydrochloric acid, and the resultant clear solution with a light olivine colour was diluted to 10 mL and adjusted to a pH of 1.5. The dye concentration was determined by comparing the solution UV–vis absorption with a standard curve of the dye.

**Substrate ingress and egress experiments**. Crystals of Zn–**TCTA** was soaked in a solution of substrate **1a** in acetonitrile (0.25 mmol/mL), and the mixture was shaken by a vortex reactor. The uptake amount of **1a** was monitored by time-course sampling of supernatant and gas chromatography (GC). The previously obtained crystals saturated with substrate **1a** was washed with a minimum amount of acetonitrile to remove the substrate absorbed on the surface, then immersed in acetonitrile, and shaken by a vortex reactor. The release amount of **1a** was monitored time-coursely by GC analysis.

**General procedure (GP) for photocatalysis by Zn–TCTA**. To a predried Pyrex tube equipped with a cooling water system, specified amounts of Zn–**TCTA**

crystals (0.00625 mmol) and substrate (0.25 mmol) were added. After adding acetonitrile (1 mL), 2,4,6-collidine (0.50 mmol), and TfCl (0.50 mmol) by syringe, the reaction mixture was stirred and illuminated with visible light by a 23 W household light under $N_2$ atmosphere for 24 h. The catalyst was recovered by centrifugation and filtration, and the filtrate was concentrated under reduced pressure. The product was isolated via flash chromatography on silica gel from the crude mixture.

**Typical procedure for ten consecutive photocatalysis runs.** To a predried Pyrex tube equipped with a cooling water system, specified amounts of photocatalyst Zn–**TCTA** (0.00625 mmol) and substrate **1a** (2.5 mmol) were added. After adding acetonitrile, 2,4,6-collidine (0.50 mmol) and TfCl (0.50 mmol) were added by syringe, and the reaction mixture was stirred and illuminated with visible light by a 23 W household light under $N_2$ atmosphere for 24 h. Then, another 0.5 mmol of additive base and TfCl were added, and the reaction was intermittently charged with the previously mentioned reagents for a total of ten times. When *fac*-Ir(Fppy)[3] was used as the photocatalyst, the reaction was performed according to the literature protocol[4], except for the use of ten equiv of **1a** (2.5 mmol) and the intermittently charged additive base and TfCl.

**Substrate encapsulation experiments.** The substrate-impregnated crystals were obtained by soaking crystals of Zn–**TCTA** in a solution of the substrate in acetonitrile (1 M) for 12 h. After the soaked Zn–**TCTA** was washed with acetonitrile, the substrate-loaded crystals were directly used for single-crystal X-ray diffraction and IR or digested with DMSO-$d^6$/DCl, and the amounts of released substrate molecules were quantified by [1]H NMR.

**$N_2$ adsorption/desorption measurements.** Low-pressure $N_2$ sorption were conducted by using a Micrometritics ASAP 2020 measurements surface area and pore size analyzer up to saturated pressure at 77 K. Before the $N_2$ sorption measurements, the fresh crystals of Zn–**TCTA** and the recovered sample were washed with acetonitrile and then subjected to vacuum heating.

## Data availability

The X-ray crystallographic coordinates for the structures reported in this article have been deposited at the Cambridge Crystallographic Data Centre (CCDC) under the deposition numbers CCDC 1407818, 1546691, 1415189, and 1415190 (Supplementary Table 1). These data can be obtained free of charge from The Cambridge Crystallographic Data Centre via http://www.ccdc.cam.ac.uk/data_request/cif. All other data supporting the findings of this study are available within the article and its Supplementary Information files or from the corresponding author upon request.

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

## Acknowledgements

This work was supported by the National Natural Science Foundation of China (21402020, U1608224, and 21531001), the Fundamental Research Funds for the Central Universities (DUT16RC(4)09 and DUT18LK50), and the 111 Project (B16008).

## Author contributions

T.Z. and X.G. contributed equally to this work. T.Z. and C.D. conceived the project, designed the experiments, and wrote the manuscript. T.Z., X.G. and Y.S. performed the experiments. C.H. and C.D. solved and refined the X-ray crystal structures. All authors discussed the results and commented on the manuscript.

## Additional information

**Competing interests:** The authors declare no competing interests.

