## [Peer Review File · Nature Communications]

Reviewers' comments:

Reviewer #1 (Remarks to the Author):

Starting from the seminal Macmillan's contribution on the application of photoredox organocatalysis to perform the trifluoromethylation of arenes, the present manuscript goes a step forward by developing a heterogeneous photocatalyst that can be recyclable. The photocatalyst is a new Zn metal organic framework, that has been characterized in the manuscript. One of the major achievements is the different regioselectivity of the trifluoromethylation observed for MOF as solid photocatalyst compared to the soluble Ir complex. This important difference has been rationalized based on the characterization by XRD of heterocycle adsorbed on the MOF. The scope of the material is broad as shown in Fig 5. The authors have finally used the MOF to promote alkene trifluoromethylation-cyclization of aryl and aroyl acrylamides. Publication in Nature Communications is recommended with some changes addressing the following comments:

1. The main point is stability of the photocatalyst. The authors indicated that the material undergoes exfoliation as determined by XRD, but typically Zn²⁺ MOFs exhibit poor stability and here this point is not well presented, particularly considering the corrosive nature of trifluorosulfonyl chloride. Analytical data of Zn in the solutions should be presented systematically.
2. Regarding the previous point, BET surface area before and after three times use must be measured and compared.
3. Comparison of powder XRD pattern of fresh and three times reused Zn-TCTA catalyst revealed that the later catalyst drastically reduced peak intensities. Hence, SEM and TEM must be shown after third cycle and compared carefully with the fresh material to demonstrate the stability of this catalyst.
4. The text and data indicate that the catalyst was recycled for three cycles but however, it contradicts with the experimental procedure. This issue must be clarified.
5. The reusability test by adding continuously reagents in the same solution is not convincing. In the way that is performed, if the solid photocatalyst dissolves and give soluble triphenylamine species that act in solution the observation could be the same. However, the solid will not be truly responsible for the process. Reuses in which the solid is filtered recovered and reused are necessary.
6. Time-conversion plots of the trifluoromethylation should be given. Actually, the best way to demonstrate the photocatalyst stability is to compare this curves upon reuse (filtration and recovery of the catalyst each time) and to show that they overlap.

Besides stability that I think that is the weakest point of the manuscript and has been improperly addressed other points are:

7. The reasons why the emission quenching presented in Fig. 2 is attributed to electron transfer quenching mechanism and the possibility that energy transfer has been ruled out are unclear. Acetone and other simple molecules are excellent energy quenchers, but here this has not been considered.
8. Authors claimed in many places that CF₃ radical is produced during the reaction, but however, there is no concrete evidence(s) to its generation under the experimental condition. The reader would have like to see some attempt to detect this radical, for instance, by trapping and EPR spectroscopy.
9. The supposedly general reaction in the top of Fig. 4 does not apply to all the examples presented in this Figure. The same applies to Fig. 5. In addition, Figs. 4 and 5 should indicate what is X and Y in each case.
10. Throughout the text the authors compare their process to enzymes. In my opinion they have gone too far. The authors have to indicate which enzyme promotes trifluoromethylation of aromatics and which utilize the light for this process. Similarly the authors have to indicate which enzyme has the periodicity in their pores like the present MOF. Otherwise, I will limit once to comment on the far similarity between the two cases.
11. Fig. 1 clarity is very poor. It is extremely difficult to read the scheme in Fig.1c. The same is also holds good for Fig. 2.
12. Figure 6 clarity is also poor.

Reviewer #2 (Remarks to the Author):

The manuscript by Duan and Co-worker reports an interesting photocatalyst system that enables the important trifluoromethylation of (hetero)arenes and tandem alkene trifluoromethylation/arene alkylation process using TfCl as trifluoromethylating reagent. This dye-based photocatalyst allows the reaction to occur under mild condition and offer good yield, thus is useful for organic synthesis. In addition, authors gave the detailed catalyst characterization, elucidating the porous structure of catalysts is essential for its catalysis performance. This catalyst could be recyclable as authors showed. These results will encourage the development of dye-based MOF photocatalysts towards practical utilization. Accordingly, I suggest accepting this manuscript for publication in Nature Communications after addressing the following questions:

- 1) Because the catalyst used here is a heterogeneous one, the size of solid particles of this catalyst should affect the catalysis efficiency. Authors need to check the effect of catalyst particle size on the reaction outcomes.
- 2) Although authors gave evidences that supported that arene substrate could be incorporated

into pores of catalyst, we cannot exclude the possibility that MOF catalyst decomposes and therefore dissolve into solution in the form molecular fragments due to coordinating acetonitrile used as a solvent. To test this hypothesis, authors should use the dye TCTA instead of Zn-TCTA MOF catalyst to check whether the reaction takes place.

3) If the reaction is homogeneous, no change for the catalyst solid isolated from reaction system is reasonable. Authors showed arene substrates is general electron-rich because of electrophilicity of CF₃ radical, which is the reason why CF₃ did not attack electron-rich catalyst dye ligand?

4) Authors are encouraged to check other electron-rich heteroarenes for this reaction, such as thiophenes, furanes for substrate scope.

Reviewer #3 (Remarks to the Author):

Duan and co-workers report the direct photocatalytic trifluoromethylation of unactivated aromatic rings at metabolically susceptible positions using a zinc-based MOF constructed using triphenylamine-based ligand consisting thiophene moieties. Series of organic compounds are synthesised using this catalyst under photochemical conditions. The as-synthesised organic compounds and the ligand precursors are well characterized by NMR techniques. However, this manuscript lacks on some fundamental issues and hence it deserves further investigations before it can be considered for publication in Nature Communications.

- One of the authors claim is that the present MOF is achieved using triphenylamine based ligand with thiophene units in a novel approach to achieve the observed results. Is it not possible to observe any activity for those MOF catalysts which can function under visible light irradiations? What is the unique in this system?
- I agree with authors that this catalyst exhibits wide substrate scope. Where does the reaction occur? What is the active site in promoting this reaction under the present experimental conditions? More experiments are required to unambiguously demonstrate reaction sites to achieve this high selectivity and activity.
- Control experiments are required to convincingly proof the high regioselectivity with MOF catalyst by comparing the activity of this MOF with analogous homogeneous catalysts under identical conditions.
- Figure 5 provides the list of substrates that have been examined under the present experimental conditions. The observed yields are ranging between 51 to 93%. What are the factors determining the differences in the yield? This issue is also valid for Figure 4.
- In-situ EPR technique along with spin-trap experiments must be provided to prove the free radical reaction pathway.
- A probable mechanism with possible experimental evidences may be proposed for this reaction

under these experimental conditions.

- The as-synthesised MOF catalyst must be characterized by BET surface area analysis for fresh and three times reused samples.
- Figures 4 and 5 may be converted into Tables with better clarity and clear details.
- Powder XRD of the three times reused sample varies compared to the fresh catalyst. This issue must be properly addressed by performing additional experiments to validate the catalyst stability.

Reviewer 1's comments:

Starting from the seminal Macmillan's contribution on the application of photoredox organocatalysis to perform the trifluoromethylation of arenes, the present manuscript goes a step forward by developing a heterogeneous photocatalyst that can be recyclable. The photocatalyst is a new Zn metal organic framework, that has been characterized in the manuscript. One of the major achievements is the different regioselectivity of the trifluoromethylation observed for MOF as solid photocatalyst compared to the soluble Ir complex. This important difference has been rationalized based on the characterization by XRD of heterocycle adsorbed on the MOF. The scope of the material is broad as shown in Fig 5. The authors have finally used the MOF to promote alkene trifluoromethylation-cyclization of aryl and aroyl acrylamides.

Publication in Nature Communications is recommended with some changes addressing the following comments:

Comment 1: The main point is stability of the photocatalyst. The authors indicated that the material undergoes exfoliation as determined by XRD, but typically Zn^{2+} MOFs exhibit poor stability and here this point is not well presented, particularly considering the corrosive nature of trifluorosulfonyl chloride. Analytical data of Zn in the solutions should be presented systematically.

Responses: Many thanks to the referee. The Zn^{2+} coordination polymer is not stable in aqueous solution, but it is relatively stable in organic solvent like acetonitrile. The organic base additive collidine could quench the HCl generated in the reaction and protect Zn^{2+} coordination polymer from the possibly existing corrosion of acidic species. The coordination polymer catalyst could be recycled for 3 times without remarkably decreased activity, showing the functional maintenance of Zn-TCTA during catalysis (Supplementary Table 2, entries 10 to 12). After each run, the supernatant of reaction mixture was subjected to inductively coupled plasma (ICP) analysis, but no zinc ions could be detected.

Comment 2: Regarding the previous point, BET surface area before and after three times use must be measured and compared.

Responses: The BET surface area analysis is of fundamental importance to examine the porous information, but we have to mention that the gas absorption/desorption processes during BET experiment make it more applicable to examine the coordination polymer pore in gas-solid biphasic processes and reactions, but not precise enough to judge the behavior of catalyst in a liquid-solid biphasic transformation of organic molecules. We performed the substrate ingress/egress experiments by comparing the fresh Zn-TCTA blocks and the recovered sample after three runs. As shown in Supplementary Fig. 12, the substrate uptaking/releasing ability of coordination polymer were not

remarkably affected by reuse. Moreover, after egress of free substrate molecules, considerable amount of **1a** molecules were still maintained in the coordination polymer pores, and the ratio of maintained **1a** to coordination polymer was calculated to be *ca.* 2.1 to 1, it was similar to the 2:1 ratio which was analyzed from the digested crystal **1a@Zn-TCTA** (Fig. 2c). Those results clearly demonstrated that the cavities of **Zn-TCTA** almostly maintained their integrity and capacity for the substrate **1a**.

Comment 3: Comparison of powder XRD pattern of fresh and three times reused Zn-TCTA catalyst revealed that the later catalyst drastically reduced peak intensities. Hence, SEM and TEM must be shown after third cycle and compared carefully with the fresh material to demonstrate the stability of this catalyst.

Responses: Many thanks to the referee for the constructive comments. The indexing of the X-ray powder diffraction patterns of the recovered catalyst indicated that the integrity of the 2D coordination polymers was maintained during the reaction and that the remarkably decreased diffraction peak at low angle corresponded to the deterioration of the long-range order in the vertical direction (Supplementary Fig. 21). On the basis of the scanning electron microscopy (SEM) (Supplementary Fig. 22) and transmission electron microscopy (TEM) images (Supplementary Fig. 23), the *in situ* exfoliation from the single-crystal blocks of 2D-stacking coordination polymer **Zn-TCTA** was observed (Fig. 4d), which should be responsible for the variant of the remarkably decreased diffraction peak at low angle of XRD spectra.

Comment 4: The text and data indicate that the catalyst was recycled for three cycles but however, it contradicts with the experimental procedure. This issue must be clarified.

Comment 5: The reusability test by adding continuously reagents in the same solution is not convincing. In the way that is performed, if the solid photocatalyst dissolves and give soluble triphenylamine species that act in solution the observation could be the same. However, the solid will not be truly responsible for the process. Reuses in which the solid is filtered recovered and reused are necessary.

(*Responses to Comments 4 and 5 will be merged together due to the logical relevancy between them.)

Responses: Many thanks to the referee for the constructive comments. As mentioned by the referee, we also performed catalyst recycle experiments for 3 times by filtering recovery and reusing. As shown in entries 10 to 12, Supplementary Table 2, **Zn-TCTA** could be isolated from the reaction mixture by centrifugation and could be reused at least three times without remarkable decrease in the reactivity. Considering the unavoidable loss when recycling a small amount of catalyst after each run, which might be responsible for the slightly decreased yield from the recycle round 1 (88% yield of **2a**), the recycle round 2 (85% yield of **2a**), to the recycle round 3 (81% yield of **2a**). Thus, we designed a time-course

experiment using a large excess of **1a** (2.5 mmol) and intermittent charging of the original amounts of TfCl and the base additive each day, after 10 consecutive runs of intermittent charging, the perfect final conversion (*ca.* 95% yield) of the time-course experiment was obtained by Zn–TCTA (Fig. 4b).

As shown in Supplementary Table 2, entries 2 and 3, either the free ligand (27% yield) or the combination of zinc salt and free ligand (44% yield) gave much lower conversions in comparison to Zn–TCTA (84% yield). Thus, it was the coordination polymer but not the solubilized free ligand should be responsible for this high conversion.

Comment 6: Time-conversion plots of the trifluoromethylation should be given. Actually, the best way to demonstrate the photocatalyst stability is to compare this curves upon reuse (filtration and recovery of the catalyst each time) and to show that they overlap.

Responses: Many thanks to the referee for the constructive comments. Time-conversion experiments by the catalysts recovered and reused for 3 cycles were performed and given in Fig. 4c, and the according superimpose of time-conversion plots well demonstrated the maintenance of reactivity of coordination polymer catalyst within 3 recycles.

Besides stability that I think that is the weakest point of the manuscript and has been improperly addressed other points are:

Comment 7: The reasons why the emission quenching presented in Fig. 2 is attributed to electron transfer quenching mechanism and the possibility that energy transfer has been ruled out are unclear. Acetone and other simple molecules are excellent energy quenchers, but here this has not been considered.

Responses: Many thanks to the referee for the constructive comments. The luminescence quenching effects of Zn–TCTA by the typical triplet quenchers acetone and 2,5-dimethylhexa-2,4-diene were examined, revealing that neither acetone nor 2,5-dimethylhexa-2,4-diene could remarkably quench the luminescence of the coordination polymer (Supplementary Fig. 11). Then, it was found that the addition of acetone (1.0 eq.) into the reaction system nearly did not deteriorate the photocatalytic result (Supplementary Table 2, entry 14). And 2,5-dimethylhexa-2,4-diene (1.0 eq.) also could not halt the photocatalytic process, although its electron rich double bonds competitively reacted with CF₃ radical, to give the formation of a reduced but still moderate yield of **2a** (Supplementary Table 2, entry 15). Thus, we thought the triplet energy transfer mechanism was not favored based upon those results.

Comment 8: Authors claimed in many places that CF₃ radical is produced during the reaction, but however, there is no concrete evidence(s) to its generation under the experimental condition. The reader would have like to see some attempt to detect this radical, for instance, by trapping and EPR spectroscopy.

Responses: Many thanks to the referee for the constructive comments. First, the negative enough reductive potential (-1.22 V) of the excited state of Zn-TCTA made it theoretically competent to undergo a SET process to reduce TfCl ($E_{1/2}^{\text{red}} = -0.18 \text{ V vs SCE}$) to generate CF_3 radical. Then, the addition of 1 equiv of typical radical scavenger tetramethylpiperidine-*N*-oxyl (TEMPO) was found to completely inhibit the reaction (Supplementary Table 2, entry 13). The possibly involved radical species in the reactions were further examined by radical intermediate trapping experiment and electron paramagnetic resonance (EPR) spectroscopy (Supplementary Fig. 24). When the reaction system of TfCl, collidine and Zn-TCTA in acetonitrile was irradiated with visible light in the presence of 2-methyl-2-nitrosopropane dimer (MNP dimer, a typical CF_3 radical scavenger), a singlet-triplet splitting signal corresponding to the $\text{CF}_3\text{-MNP}\cdot$ adduct was observed ($g = 2.006$), confirming the formation of CF_3 radicals during the trifluoromethylation of arenes.

Comment 9: The supposedly general reaction in the top of Fig. 4 does not apply to all the examples presented in this Figure. The same applies to Fig. 5. In addition, Figs. 4 and 5 should indicate what is X and Y in each case.

Responses: Many thanks to the referee for the constructive comments. We have revised the due sections as suggested by the referee.

Comment 10: Throughout the text the authors compare their process to enzymes. In my opinion they have gone too far. The authors have to indicate which enzyme promotes trifluoromethylation of aromatics and which utilize the light for this process. Similarly the authors have to indicate which enzyme has three periodicity in their pores like the present MOF. Otherwise, I will limit once to comment on the far similarity between the two cases.

Responses: We apologized for misleading the referee by the over-description in the text, and the relative sections in the paper were revised based upon the comments of referee.

Comment 11: Fig. 1 clarity is very poor. It is extremely difficult to read the scheme in Fig.1c. The same is also holds good for Fig. 2.

Responses: Sorry for the poor clarities of figures, and we have replaced them with ones of high enough dpi.

Comment 12: Figure 6 clarity is also poor.

Responses: We apologize for the low quality of figures, thus we replaced Fig. 6 with a better one as referee mentioned.

Reviewer 2's comments:

Publication in Nature Communications is recommended with some changes addressing the following comments:

The manuscript by Duan and Co-worker reports an interesting photocatalyst system that enables the important trifluoromethylation of (hetero)arenes and tandem alkene trifluoromethylation/arene alkylation process using TfCl as trifluoromethylating reagent. This dye-based photocatalyst allows the reaction to occur under mild condition and offer good yield, thus is useful for organic synthesis. In addition, authors gave the detailed catalyst characterization, elucidating the porous structure of catalysts is essential for its catalysis performance. This catalyst could be recyclable as authors showed. These results will encourage the development of dye-based MOF photocatalysts towards practical utilization. Accordingly, I suggest accepting this manuscript for publication in Nature Communications after addressing the following questions:

Comment 1: Because the catalyst used here is a heterogeneous one, the size of solid particles of this catalyst should affect the catalysis efficiency. Authors need to check the effect of catalyst particle size on the reaction outcomes.

Responses: Many thanks to the referee for the constructive comments. The catalytic performance of grounded Zn-TCTA with decreased particle size (as shown in Supplementary Fig. 22d) was examined, there was no remarkable difference in catalytic efficiency compared with the case using ungrounded coordination polymer blocks in general procedure (Supplementary Table 2, entries 1 and 16).

Comment 2: Although authors gave evidences that supported that arene substrate could be incorporated into pores of catalyst, we cannot exclude the possibility that MOF catalyst decomposes and therefore dissolve into solution in the form molecular fragments due to coordinating acetonitrile used as a solvent. To test this hypothesis, authors should use the dye TCTA instead of Zn-TCTA MOF catalyst to check whether the reaction takes place.

Responses: Many thanks to the referee for the constructive comments. The free rotation of phenyl blades of triphenylamine moiety containing free ligand H₃TCTA resulted in the consumption of harvested light energy in the form of non-radiative pathways, while the incorporation of TCTA into coordination polymer restricted its free rotation and remarkably increase the luminescence efficiency and benefit the photocatalysis by coordination polymer. As shown in Supplementary Table 2, entries 2 and 3, neither the free ligand (27% yield) nor the combination of zinc salt and free ligand (44% yield) would give comparable conversion as high as Zn-TCTA (84% yield), which obviously did not support the hypothesis that the decomposed coordination polymer dominated the reaction.

Comment 3: If the reaction is homogeneous, no change for the catalyst solid isolated from reaction system is reasonable. Authors showed arene substrates is general electron-rich because of electrophilicity of CF₃ radical, which is the reason why CF₃ did not attack electron-rich catalyst dye ligand?

Responses: Many thanks to the referee for the constructive comments. We digested the coordination polymer samples recovered from the reaction mixture to check the possible incorporation of fluoro containing groups on the scaffold, however, no remarkable trifluoromethylation products could be detected from the ¹⁹F NMR. Although the thiophene moiety increase the electron density of ligand, the existence of electron withdrawing carboxylic group and the coordination of Lewis acidic zinc ion remarkably decreased the overall electron density of ligand, moreover, the nitrogen centered cationic radical intermediate during photoredox cycle further enhanced the electron deficient degree. Those factors didn't benefit the involvement of ligand in the trifluoromethylation. On the other hand, in comparison to the catalytic amount of Zn-TCTA, the large excess amount of aromatic substrates were more reactive towards the CF₃ radical. Moreover, it was proven that the substrate could enter the channels of coordination polymer and be "docked" in the intralayer pores rapidly, which ingress rate (only *ca.* 5 min was needed to reach the saturated absorption in the case of **1a**) was much more efficient than the reaction rate (24 hrs was needed for the completion of reaction). Thus, either from a thermodynamic or kinetic point of view, it was believed that the trifluoromethylation of TCTA moiety of coordination polymer was not favored.

Comment 4: Authors are encouraged to check other electron-rich heteroarenes for this reaction, such as thiophenes, furanes for substrate scope.

Responses: Many thanks to the referee for the constructive comments. We have examined the reactions using electron-rich thiophene (84 °C) or furan (b.p. 31.3 °C) as substrates, but the serious loss of products due to the low boiling points of corresponding trifluoromethylated products (such as 2-trifluoromethyl-thiophene b.p. 60 °C) made the isolation and purification process quite challenge to handle. In the aim of practical application, the electron-rich benzofuran and benzothiophene derivatives with higher boiling points were checked, and the corresponding trifluoromethylated products were isolated in good yields (Table 1, **2g** and **2h**).

Reviewer 3's comments:

Duan and co-workers report the direct photocatalytic trifluoromethylation of unactivated aromatic rings at metabolically susceptible positions using a zinc-based MOF constructed using triphenylamine-based ligand consisting thiophene moieties. Series of organic compounds are synthesised using this catalyst under photochemical conditions. The as-synthesised organic compounds and the ligand precursors are well characterized by NMR techniques. However, this manuscript lacks on some fundamental issues and hence it deserves further investigations before it can be considered for publication in Nature Communications.

Comment 1: One of the authors claim is that the present MOF is achieved using triphenylamine based ligand with thiophene units in a novel approach to achieve the observed results. Is it not possible to observe any activity for those MOF catalysts which can function under visible light irradiations? What is the unique in this system?

Responses: Many thanks to the referee for the enlightening comments. As depicted by Prof. Cohen (*J. Am. Chem. Soc.* **138**, 12320–12323 (2016)), the homogeneous photocatalytic Ir(III) polypyridyl complexes could be immobilized to the famous zirconium MOF UiO-67-bpy, and the obtained UiO-67-Ir(ppyF)₂ was proven to be competent in photocatalysis under visible light irradiation. When the Ir(III) containing UiO-MOF photocatalyst was examined in the photocatalytic trifluoromethylation of **1a**, a mixture of regioisomers and over-trifluoromethylated products was obtained (Supplementary Table 2, entry 17). The famous triphenylamine based MOF-150, which was amenable of photoreductive generation of alkyl radical upon visible light irradiation (*J. Am. Chem. Soc.* **134**, 14991–14999 (2012)), was also examined here. When electron rich substrate **1j** was used, remarkable amounts of over-trifluoromethylated and non-photoirradiative side products were detected (Supplementary Fig. 20) in the presence of MOF-150. Obviously, the milder generation of CF₃ radicals, the matched reaction dynamic control, the good light transmittance and mass transfer ability are important factors for heterogeneous photocatalytic system to achieve good efficiency and selectivity.

In the case of Zn-TCTA, the insertion of thiophene moieties into the backbone of a TPA-based ligand extended the visible light absorption ability and tuned the photoelectronic properties, decreasing the harvested photon energy in comparison to TPA core to ensure the milder generation of CF₃ radicals and the enhanced ability to oxidize the CF₃ radical addition intermediate. The enlarged π -system of the ligand favored the formation of 2D sheets and caused the metal nodes to be distorted, creating coordination vacancies to dock the substrates near the photocatalytically active centers and even stabilizing unique conformations of the substrates. The comprehensive improvements within this unique system using Zn-TCTA well-balanced the contradictory demands of the electronic effects and the reaction dynamics,

achieving regio- and diastereoselective discrimination between the reaction sites with unremarkable electronic/steric differences.

Comment 2: I agree with authors that this catalyst exhibits wide substrate scope. Where does the reaction occur? What is the active site in promoting this reaction under the present experimental conditions? More experiments are required to unambiguously demonstrate reaction sites to achieve this high selectivity and activity.

Responses: Thanks to the referee for the constructive comments. It was believed that the reaction mainly occurred in the pore of Zn-TCTA. The substrate ingress/egress experiments showed that indomethacin methyl ester **1o** (with a diameter of *ca.* 13.1 Å) could be uptaken by Zn-TCTA (with a cross-section of 10.4 × 18.0 Å² open channel) (Supplementary Fig. 13), but the oversized substrate indomethacin *n*-nonylphenyl ester **1p** (with a diameter of *ca.* 22.9 Å) could not be encapsulated within the pore of coordination polymer. Not surprisingly, in the presence of Zn-TCTA, the conversion of reaction using **1p** (<10% yield) were much lower than the one using **1o** (58% yield). Another clue was that the efficiency of reaction had obviously positive correlation with the rate of substrate diffusion within Zn-TCTA. In the case of Table 1, **2a** (84% yield), the small-sized substrate **1a** could diffuse into the pores of coordination polymer and reach the saturation within less than 10 min (Fig. 4a and Supplementary Fig. 12). However, in the case of Table 1, **2o** which obtained a moderate 58% yield, the ingress and egress of much bigger-sized substrate **1o** within Zn-TCTA were at a much slower rate compared with **1a**, and nearly as long as 10 hrs would be needed to reach the absorption saturation (Supplementary Fig. 13). The diffusion limitation of **1o** within coordination polymer seemingly retarded the further conversion. Thus, the pore of Zn-TCTA was considered as the “active site” to achieve the promoted activity.

On the other hand, based upon the comparative study on catalytic performance of heterogeneous Zn-TCTA and homogeneous H₃TCTA and *fac*-Ir(Fppy)₃ (Supplementary Fig. 17), we could see that the homogeneous photocatalytic system gave inferior regioselectivities, which also reflected the fundamental importance of confined environment of pores within Zn-TCTA for the improved selectivity.

Comment 3: Control experiments are required to convincingly proof the high regioselectivity with MOF catalyst by comparing the activity of this MOF with analogous homogeneous catalysts under identical conditions.

Responses: As suggested by the referee, the control experiments were performed by comparing the coordination polymer with the free ligand H₃TCTA under identical conditions (Supplementary Fig. 17). Obviously, the homogeneous counterpart H₃TCTA gave much lower conversions and inferior regioselectivities in comparison to Zn-TCTA, reflecting the importance of local confined environment of

coordination polymer.

Comment 4: Figure 5 provides the list of substrates that have been examined under the present experimental conditions. The observed yields are ranging between 51 to 93%. What are the factors determining the differences in the yield? This issue is also valid for Figure 4.

Responses: Many thanks to the referee for the enlightening comments. The final conversion of a reaction here involves various factors, at least including the steric and electronic effects of substrate, and the confined environments within the coordination polymer. Thus, the variant yields in Figs. 4 and 5 (*now labeled as Tables 1 and 2) will be rationalized and categorized in a case by case manner.

1) **Electronic effect of substrate:** Generally, the highly electrophilic CF_3 radical favors the electron rich substrate to give higher conversions. In Table 1, **2f**, the existence of lactone ($-\text{CO}_2-$) decreased the electron density of coumarin **1f**, giving a moderate conversion of 62%.

2) **Accessibility of reaction site of substrates to the active site within pores:** This is a comprehensive result of steric effect of substrates and the confinement effect of pores of coordination polymer.

In comparison to the methyl neighboring group of reaction site of Table 1, **1d**, the more steric chloroethyl neighboring group in the case of **1e** didn't benefit the accessibility of reaction site of **1e** towards the photocatalytic site of Zn-TCTA, moreover, the inherent confined environment of coordination polymer pores might aggravate the steric effect, thus a much lower yield was not surprising (41% of **2e** vs. 87% of **2d**). Similarly, in comparison to the cases of **4f** (69% yield, Table 2) and **4g** (67% yield, Table 2) bearing less bulky vicinal dimethyl groups on unsaturated alkene moieties, the substrates with more steric vicinal rings (53% and 57% yields, Table 2, **4h** and **4i**) or the one with bulky Boc group (51% yield, Fig. 5, **4j**), lower conversions were observed.

On the other hand, in the cases of Table 1, **2o** (58% yield), although there was not remarkable steric hindrance around the possible reaction sites of substrate **1o**, but the limited diffusion of big-sized and rigid substrates within confined environment of coordination polymer (Supplementary Fig. 13) hampered the efficiency of reaction, and simply lengthening the reaction time of Table 1, **2o** could not effectively increase the yield. As expected, enlarging the substrate size to exceed the diameter of open channel of coordination polymer inhibited the accessibility of over-sized substrate **1p** to the photocatalytic site within pores (Supplementary Fig. 13), leading to sharply decreased conversion (Table 1, **2p**, <10% yield).

Comment 5: In-situ EPR technique along with spin-trap experiments must be provided to prove the free radical reaction pathway.

Responses: Many thanks to the referee for the constructive comments. The addition of 1 equiv of typical

radical scavenger tetramethylpiperidine-*N*-oxyl (TEMPO) completely inhibited the reaction (Supplementary Table 2, entry 13). And of the reaction mixture with 2-methyl-2-nitrosopropane dimer (MNP dimer, a typical CF₃ radical scavenger) to substitute the substrate, a singlet–triplet splitting signal corresponding to the CF₃–MNP• adduct was observed by electron paramagnetic resonance (EPR) spectroscopy (*g* value = 2.006) under light irradiation. These results confirmed the formation of CF₃ radicals during this photocatalysis (Supplementary Fig. 24), which confirms the formation of CF₃ radicals during the trifluoromethylation of arenes.

Comment 6: A probable mechanism with possible experimental evidences may be proposed for this reaction under these experimental conditions.

Responses: Based upon the mechanistic description by MacMillan (*Nature* **480**, 224–228 (2011).) and our experimental results, a plausible mechanism was proposed here by taking the case of Table 1, **2a** as an example. The substrate rapidly diffuses in the pore of Zn-TCTA (Fig. 4a and Supplementary Fig. 12) and is “docked” near to the photocatalytic center in an early transition state that the possible reaction sites of substrate are endowed with differentiated spatial proximities towards the photocatalytic center (Fig. 3d). The excited state Zn-TCTA undergoes a single-electron transfer (SET) to triflyl chloride (TfCl, CF₃SO₂Cl) (Fig. 2b) with concurrent oxidation of ligand nitrogen center to the cationic radical form. Then the generated CF₃SO₂Cl radical anion rapidly collapses to give the CF₃ radical (Supplementary Fig. 24), which is entropically driven by the release of SO₂ and chloride anion. Then the highly electrophilic CF₃ radical adds to one of several electron-rich positions of **1a** which has shorter spatial proximity towards the photocatalytic center, to give the dearomatized carbon centered radical intermediate (Fig. 3e), which is then subjected to a SET oxidation event by the nitrogen centered cationic radical to give carbon centered cation intermediate (Fig. 3f) and regenerate ground-state photocatalyst. Finally, the carbon centered cation undergoes deprotonation assisted by the organic base collidine, providing the desired trifluoromethylated aromatic ring **2a**.

Comment 7: The as-synthesised MOF catalyst must be characterized by BET surface area analysis for fresh and three times reused samples.

Responses: The BET surface area analysis is of fundamental importance to examine the porous information, but we have to mention that the gas absorption/desorption processes during BET experiment make it more applicable to examine the coordination polymer pore in gas-solid biphasic processes and reactions, but not precise enough to judge the behavior of catalyst in a liquid-solid biphasic transformation of organic molecules. We performed the substrate ingress/egress experiments by comparing the fresh Zn-TCTA blocks and the recovered sample after three runs. As shown in Supplementary Fig. 12, the substrate uptaking/releasing ability of coordination polymer were not

remarkably affected by reuse. Moreover, after egress of free substrate molecules, considerable amount of **1a** molecules were still maintained in the coordination polymer pores, and the ratio of maintained **1a** to coordination polymer was calculated to be *ca.* 2.1 to 1, it was similar to the 2:1 ratio which was analyzed from the digested crystal **1a@Zn-TCTA** (Fig. 2c). Those results clearly demonstrated that the cavities of **Zn-TCTA** almostly maintained their integrity and capacity for the substrate **1a**.

Comment 8: Figures 4 and 5 may be converted into Tables with better clarity and clear details.

Responses: Many thanks to the referee for the constructive comments. Figures 4 and 5 have been converted into Table 1 and 2, respectively, with better clarity.

Comment 9: Powder XRD of the three times reused sample varies compared to the fresh catalyst. This issue must be properly addressed by performing additional experiments to validate the catalyst stability.

Responses: Many thanks to the referee for the constructive comments. The indexing of the X-ray powder diffraction patterns of the recovered catalyst indicated that the integrity of the 2D coordination polymers was maintained during the reaction and that the remarkably decreased diffraction peak at low angle corresponded to the deterioration of the long-range order in the vertical direction (Supplementary Fig. 21). On the basis of the scanning electron microscopy (SEM) (Supplementary Fig. 22) and transmission electron microscopy (TEM) images (Supplementary Fig. 23), the *in situ* exfoliation from the single-crystal blocks of 2D-stacking coordination polymer **Zn-TCTA** was observed (Fig. 4d), which should be responsible for the variant of the remarkably decreased diffraction peak at low angle of XRD spectra. Thus, we performed a substrate adsorption/release experiment to check the pores of coordination polymer before and after the catalysis. As shown in Supplementary Fig. 12, the capacity of pores of **Zn-TCTA** towards the substrate **1a** almostly maintained after 3 cycles of reuse, demonstrating that the active sites of coordination polymer still kept their integrity and function.

Reviewers' comments:

Reviewer #1 (Remarks to the Author):

The authors have satisfactorily addressed most of my previous comments, particularly those related to MOF stability that is against the general poor stability of Zn²⁺ MOFs. They have also provided evidence of the generation of CF₃· radicals, making stronger their mechanistic proposal. I have had also the opportunity to read the reports of the other two reviewer's and the author's responses. Thus, I think this submission should be accepted for publication. I would ask only for some minor changes addressing the following two points that were common for me and reviewer 3 and not yet fully solved:

BET surface area data of the fresh and used catalyst should be given. This is mandatory since it will report on the porosity of the material and complement the liquid-phase adsorption measurements made already by the authors.

Related to the previous point and concerning the ingress/egress measurements indicated by the authors, if I understand well, it was not possible to recover completely compound 1a in the desorption extractions. Is this correct? Then, when performing the preparative experiments, are also part of substrates/products unrecoverable? Has this been sufficiently highlighted and indicated in the text? How can the photocatalyst be stable if part of the material remains adsorbed and is unrecoverable? If the reaction is occurring inside the pores as claimed, how can be the photocatalyst stable and not decrease in activity if the pores become clogged? I suppose that the authors have not properly explained their adsorption/desorption measurements, since otherwise the whole paper has to be re-revised.

Reviewer #2 (Remarks to the Author):

During revision of this manuscript, Authors have seriously addressed my concerns based on their further experiments. Right now, authors have answered a series of questions as for whether the size of heterogeneous catalyst influence the outcomes of the catalytic reaction, stability of MOF catalyst exposed to photodriven reaction conditions, and whether this catalysis process is heterogeneous or homogeneous. in response to my suggestion, authors have expand the substrate scope to heteroarenes. thus, the current state of this manuscript meets the criteria for publishing in Nature Communications. I suggest accepting this manuscript as it is.

Reviewer #3 (Remarks to the Author):

Manuscript ID: NCOMMS-17-34510A

Authors have satisfactorily addressed all the three reviewer's comments and the original version has been revised appropriately. Hence, the above referenced revised version may now be recommended for publication in Nature Communications.

Reviewer 1's comments:

The authors have satisfactorily addressed most of my previous comments, particularly those related to MOF stability that is against the general poor stability of Zn^{2+} MOFs. They have also provided evidence of the generation of $CF_3\cdot$ radicals, making stronger their mechanistic proposal. I have had also the opportunity to read the reports of the other two reviewer's and the author's responses. Thus, I think this submission should be accepted for publication. I would ask only for some minor changes addressing the following two points that were common for me and reviewer 3 and not yet fully solved:

Comment 1: BET surface area data of the fresh and used catalyst should be given. This is mandatory since it will report on the porosity of the material and complement the liquid-phase adsorption measurements made already by the authors.

Responses: Many thanks to the suggestive comments of referee. Low-pressure N_2 adsorption/desorption of fresh Zn-TCTA and the recycled sample were conducted at 77 K. As shown in Supplementary Fig. 28, both of fresh crystals of Zn-TCTA and the recovered sample exhibited a Type-I sorption behavior, with Brunauer-Emmett-Teller (BET) surface areas of 1383 and 1600 $m^2 g^{-1}$, respectively, which were calculated from N_2 adsorption isotherms. Compared with the case of fresh Zn-TCTA, the mildly increased BET surface area of recovered sample might be attributed to the *in situ* exfoliation. The maintained pore size distributions reflected the stability of coordination polymer and the integrity of pores during photocatalysis.

Comment 2: Related to the previous point and concerning the ingress/egress measurements indicated by the authors, if I understand well, it was not possible to recover completely compound 1a in the desorption extractions. Is this correct? Then, when performing the preparative experiments, are also part of substrates/products unrecoverable? Has this been sufficiently highlighted and indicated in the text? How can the photocatalyst be stable if part of the material remains adsorbed and is unrecoverable? If the reaction is occurring inside the pores as claimed, how can be the photocatalyst stable and not decrease in activity if the pores become clogged? I suppose that the authors have not properly explained their adsorption/desorption measurements, since otherwise the whole paper has to be re-revised.

Responses: Many thanks to the enlightening comments of referee. Yes, we agreed that some amount of substrate 1a was retained in the pores of Zn-TCTA after the desorption extraction. It should be noted that the demand of adsorption/desorption in the field of catalysis is different from that in the case of gas adsorption/desorption, and the retaining of substrate in pores should be important for the smooth conversion of photocatalysis. As shown in Fig. 4a and Supplementary Fig. 15, in comparison to the rapid adsorption of substrate 1a and desorption of product 2a, the photocatalytic reaction is the

rate-determining step. Importantly, the ingress and egress experiment of product **2a** didn't shown remarkable amount of retained **2a** in Zn-TCTA after the desorption extraction (Supplementary Fig. 15). Moreover, the substrate-product exchange experiment demonstrated that after reaching the saturated adsorption of product **2a**, the Zn-TCTA crystals could still remarkably adsorb the substrate **1a**, and product **2a** was expelled from pores effectively in this competitive sorption process (Supplementary Fig. 17). After appending an electron withdrawing CF₃ group, the coordination ability of carbonyl of product **2a** is weaker compared with that of substrate **1a**, which might be responsible for the easier diffusion out behavior of the product. When performing the reaction using the stoichiometric amount of Zn-TCTA crystals encapsulated with substrate **1a** instead of the separately added Zn-TCTA and substrate, the reaction finished under 1 h of visible-light irradiation to give the formation of product **2a** in 95% isolated yield (Supplementary Table 2, entry 18), and ¹H-NMR of digested sample of recovered catalyst didn't shown peaks of substrate **1a** or product **2a**. Similarly, after the preparative experiments, the retaining of substrates/products within catalysts were not observed. Furthermore, N₂ adsorption isotherms and related analysis demonstrated the stability of catalyst Zn-TCTA, and the characteristics of pores were maintained during photocatalysis (Supplementary Fig. 28).

Reviewer 2's comments:

During revision of this manuscript, Authors have seriously addressed my concerns based on their further experiments. Right now, authors have answered a series of questions as for whether the size of heterogeneous catalyst influence the outcomes of the catalytic reaction, stability of MOF catalyst exposed to photodriven reaction conditions, and whether this catalysis process is heterogeneous or homogeneous. In response to my suggestion, authors have expand the substrate scope to heteroarenes. Thus, the current state of this manuscript meets the criteria for publishing in Nature Communications. I suggest accepting this manuscript as it is.

Responses: Many thanks to the referee for the positive comments.

Reviewer 3's comments:

Authors have satisfactorily addressed all the three reviewer's comments and the original version has been revised appropriately. Hence, the above referenced revised version may now be recommended for publication in Nature Communications.

Responses: Many thanks to the referee for the positive comments.